

# Reanalysis intercomparison of potential vorticity and potential-vorticity-based diagnostics

Luis F. Millán[1], Gloria L. Manney[2,3], and Zachary D. Lawrence[4,5,6]

[1]Jet Propulsion Laboratory, California Institute of Technology, Pasadena, California, USA
[2]NorthWest Research Associates, Socorro, New Mexico, USA
[3]New Mexico Institute of Mining and Technology, Socorro, New Mexico, USA
[4]Cooperative Institute for Research in Environmental Sciences (CIRES), University of Colorado, Boulder Colorado, USA
[5]NOAA Physical Sciences Laboratory (PSL), Boulder, Colorado, USA
[6]NorthWest Research Associates, Socorro, New Mexico, USA

**Correspondence:** L. Millán (luis.f.millan@jpl.nasa.gov)

**Abstract.** Global reanalyses from data assimilation systems are among the most widely used datasets in weather and climate studies, and potential vorticity (PV) from reanalyses is invaluable for many studies of dynamical and transport processes. We assess how consistently modern reanalyses represent potential vorticity (PV) among each other, focusing not only on PV but also on process-oriented dynamical diagnostics including equivalent latitude calculated from PV and PV-based tropopause and stratospheric polar vortex characterization.

In particular we assess the National Centers for Environmental Prediction Climate Forecast System Reanalysis/Climate Forecast System, version 2 (CFSR/CFSv2) reanalysis, the European Centre for Medium-Range Weather Forecasts Interim (ERA-Interim) reanalysis, the Japanese Meteorological Agency's 55-year (JRA-55) reanalysis, and the Modern-Era Retrospective analysis for Research and Applications, version 2 (MERRA-2). Overall, PV from all reanalyses agrees well with the reanalysis ensemble mean, providing some confidence that all of these recent reanalyses are suitable for most studies using PV-based diagnostics. Specific diagnostics where some larger differences are seen include PV-based tropopause locations in regions that have strong tropopause gradients (such as around the subtropical jets) or are sparse in high-resolution data (such as over Antarctica), and the stratospheric polar vortices during fall vortex formation and (especially) spring vortex breakup; studies of sensitive situations / regions such as these should examine PV from multiple reanalyses.

## 1 Introduction

Global reanalyses provide gridded high-resolution meteorological fields over several decades based on an optimized combination of general circulation models and observational data. That is, data assimilation methods ingest observations to constrain the general circulation models and provide spatially and temporally consistent atmospheric states, offering a wide range of





variables, such as temperature, humidity, winds, and vorticity. They are among the most widely used datasets in the study of weather and climate.

As part of the Stratosphere–troposphere Processes and their Role in Climate (SPARC) Reanalysis Intercomparison Project (S-RIP), we assess how consistently reanalyses represent potential vorticity (PV) among each other. As described by Hoskins
et al. (1985), PV can be expressed as the product of vorticity (the local rate of rotation of the air parcels) and static stability (the gravitational resistance of the atmosphere to vertical displacements). PV is conserved in adiabatic, frictionless flow, and since this type of flow approximates many flows in the real atmosphere, PV acts approximately as a tracer of the movement of air parcels (e.g., McIntyre and Palmer, 1984; Hoskins et al., 1985). Thus, PV has been commonly used as a horizontal spatial coordinate, especially for stratospheric studies where very large scales of motion dominate. That is, on isentropic surfaces,
PV is frequently sufficiently monotonic in latitude to act as a dynamical coordinate, typically scaled to PV equivalent latitude, which is valuable for identifying transport barriers and studying how they affect trace gas motions (e.g., Norton, 1994; Lary et al., 1995; Manney et al., 1999; Allen and Nakamura, 2003). Further, not only can PV be viewed as a tracer for many purposes, but also the complete flow structure (balanced winds and temperature) can be determined from the spatial distribution of PV itself (a property referred as the invertibility principle, e.g., Hoskins et al., 1985).

These properties of PV make it useful for identifying features of the atmosphere. For example, PV has been extensively used to identify the location of the tropopause (e.g., Schoeberl, 2004; Manney et al., 2007; Kunz et al., 2011) since the stratosphere possesses higher values of PV than the troposphere and changes in static stability result in strong vertical PV gradients at the tropopause (e.g., Morgan and Nielsen-Gammon, 1998). Determining the tropopause accurately is important because it acts as a dynamic barrier, dividing the troposphere where trace gases tend to be well mixed and the stratosphere where they tend
to be strongly stratified and thus have strong vertical gradients. Similarly, the boundary of the stratospheric polar vortex is typically characterized by steep PV gradients (e.g., McIntyre and Palmer, 1983; Clough et al., 1985; Nash et al., 1996), PV fields have been extensively used to identify such vortices (e.g., Hoskins et al., 1985; Waugh and Randel, 1999; Manney et al., 2007). Because the strong PV gradients along the vortex edge act as a transport barrier, the air masses inside and outside are often substantially different, with the former physically isolated from the rest of the atmosphere, sometimes for many
25  months. Determining the vortex edge is thus critical for studies of polar ozone loss and chemical processing, and for studies of stratospheric transport and mixing processes (e.g., Butchart and Remsberg, 1986; Leovy et al., 1985; Manney et al., 2009; Manney and Lawrence, 2016, and references therein).

Despite its importance and many uses, comparisons of PV in recent reanalyses are limited to a few focused comparisons for specific processes (e.g., maximum PV gradients representing the vortex edge in Lawrence et al., 2018). In this study,
we intercompare the European Centre for Medium-Range Weather Forecasts (ECMWF) ERA-Interim reanalysis, the National Centers for Environmental Prediction Climate Forecast System Reanalysis and Climate Forecast System Version 2 (CFSR/CFSv2), the Japanese 55-year Reanalysis (JRA-55), and the Modern-Era Retrospective Analysis for Research and Applications-2 (MERRA-2). Section 2 briefly describes the reanalyses used as well as the calculations, if any, to compute PV for those reanalyses. Section 3 identifies discontinuities in those PV fields related to changes in the assimilated data and
processing streams. Section 4 intercompares the reanalysis PV fields and section 5 intercompares PV-based equivalent latitude.





Lastly, sections 6 and 7 evaluate the impact of the PV differences upon dynamical tropopauses and vortex characterization, respectively.

## 2 Reanalyses and Methods

As mentioned, the reanalyses used in this study are MERRA-2 (Gelaro et al., 2017), ERA-Interim (Dee et al., 2011), CFSR/CFSv2
(Saha et al., 2010, 2014) and JRA-55 (Kobayashi et al., 2015). A detailed overview of the models, assimilation schemes, and assimilated data is given by Fujiwara et al. (2017). Table 1 summarizes the horizontal and vertical grids and lid heights for these reanalyses. Throughout this study we use the 12 UT synoptic fields for all days from 1980 through 2014, starting with the fields on the native model levels and horizontal grids (or, for spectral models, regular latitude/longitude grids with resolution near that of the Gaussian grid associated with the spectral resolution).

Of these reanalyses, only MERRA-2 provides PV calculated within the assimilation system on the model grid. CFSR/CFSv2 provides relative vorticity while ERA-Interim provides absolute vorticity, hence, for CFSR/CFSv2 and ERA-Interim we estimate PV from the reanalyses temperature, pressure, and their provided vorticity. For JRA-55, since neither absolute nor relative vorticity is provided on model levels, PV is estimated directly from the reanalysis winds and temperatures (interpolated to isentropic surfaces) using a method similar to the ones described by Newman et al. (1989) and Manney and Zurek (1993). PV on
an isentropic surface is given by,

$$PV = -g(\varsigma_\theta + f)\frac{\partial \theta}{\partial p} \tag{1}$$

where $g$ is gravity, $f$ is the Coriolis parameter (the planetary vorticity), $p$ is pressure, $\theta$ is potential temperature, and $\varsigma_\theta$ is the component of relative vorticity orthogonal to the $\theta$ surface. As mentioned above, for JRA-55, relative vorticity is estimated from the zonal and meridional reanalysis winds. Overall, because of the Coriolis parameter, PV increases monotonically from
20 negative values at the South Pole to positive values at the North Pole.

We scale the PV fields to vorticity units as in Dunkerton and Delisi (1986), that is,

$$sPV = \frac{PV}{g|\partial\theta_o/\partial p|} \tag{2}$$

where $\partial\theta_o/\partial p$ is calculated assuming a constant lapse rate of $1\,\mathrm{K\,km^{-1}}$ and a pressure of $54\,\mathrm{hPa}$ at $500\,\mathrm{K}$ isentropic surface (Manney et al., 1994). This scaling is performed to provide fields with a similar order of magnitude throughout the stratosphere
as opposed to PV (which increases approximately exponentially with increasing $\theta$). Further, since all reanalyses are on different vertical and horizontal grids, the sPV fields were interpolated to a set fix of potential temperatures, and onto a $0.5°$ by $0.5°$ horizontal grid. To provide an unbiased view, a reanalysis ensemble mean (REM) is used in this study as a comparison tool. This REM is not meant to be considered as "truth" but simply as a baseline to identify similarities and differences among the reanalysis.



## 3 Reanalysis related sPV discontinuities

Even though reanalyses use a frozen configuration for modeling and data assimilation procedures to produce the most homogeneous set of fields as possible, the observations available to assimilate vary over time, which can introduce discontinuities in reanalysis fields; further, discontinuities can also be introduced by breaking the record into multiple processing streams (see, e.g., Fujiwara et al., 2017, for descriptions of observational input changes and processing streams). Figure 1 displays sPV anomaly time series for each reanalysis from their corresponding monthly climatological values for 90°S to 60°S. That is, the anomalies for ERA-Interim are computed with respect to the ERA-Interim climatology, the MERRA-2 anomalies are computed with respect to the MERRA-2 climatology, and so on. These anomalies help identify sudden discontinuities (abrupt changes) that may be caused by changes in assimilated datasets or between processing streams. More gradual changes (due to real trends for example) may also appear in these anomaly plots but generally will not coincide with documented observing system or processing changes. We choose this region because it highlights the anomalies slightly better than other latitude bins, presumably due to the scarcity of data in the south polar regions; however, similar discontinuities are seen in other latitude bins.

The most obvious discontinuity in most reanalyses occurs at the time of the transition between the TIROS Operational Vertical Sounder (TOVS) and the Advanced TOVS (ATOVS) suites in October 1998. The impact of this transition upon reanalysis fields results from the improvement in vertical resolution of radiances from the advanced suite and it is a well documented feature (e.g., Onogi et al., 2007; Fujiwara et al., 2017; Long et al., 2017; Lawrence et al., 2018). This PV discontinuity appears much more pronounced in MERRA-2 and ERA-Interim than in CFSR/CFSR-2 or JRA-55, similar to the results of Lawrence et al. (2018) for polar processing diagnostics. ERA-Interim shows a discontinuity in 1985 resulting from the transition from the NOAA-7 to the NOAA-9 Stratospheric Sounding Unit (SSU) (Simmons et al., 2014). The CFSR discontinuities in 2008 and 2011, related to the start of assimilation of Infrared Atmospheric Sounding Interferometer (IASI) radiances and the start of the new assimilation system (CFSv2) (Saha et al., 2014), respectively, are the most pronounced discontinuities in the CFSR/CFSv2 record. MERRA-2 discontinuities in late 1994 and 2004 arise from changes in the assimilation of the Solar Backscatter Ultraviolet Radiometer (SBUV) and from the start of assimilation of Microwave Limb Sounder (MLS) temperatures (Wargan et al., 2017), respectively. The impact of these discontinuities will be discussed further in the following sections.

## 4 Variations from climatology

Figure 2 shows the sPV zonal mean seasonal climatology (1980 - 2014). Peak sPV values are seen in each hemisphere's winter, that is, maxima around DJF in the Northern Hemisphere and JJA in the Southern Hemisphere. This season also shows the largest variability in each hemisphere.

Figure 2 also shows the sPV differences between the reanalysis fields and the REM along with overlaid contours for each reanalyses' sPV climatology. Overall, all reanalyses agree with the REM within $0.1*10^{-4}$ s$^{-1}$, but there are some biases. The most pronounced differences are for CFSR/CFSv2 during each hemisphere's winter (and to a lesser degree during fall), with the magnitude of sPV biased low by up to $1*10^{-4}$ s$^{-1}$ in each hemisphere (i.e., up to around a 40% bias at 2500 K).





To further investigate the impact of the discontinuities identified and briefly discussed in section 3 (that is, discontinuities that could be related to different assimilated datasets or different processing streams), in Figure 3 we examine sPV yearly differences with respect to the yearly REM. Further, this figure allows us to study how the agreement among the reanalyses changes with time. Figure 3 shows the sPV differences for each reanalysis from the REM at 70°S in JJA and at 70°N in DJF.
These periods and latitudes are chosen to show differences in each hemisphere where (1) the most sPV variability is found and (2) the most pronounced differences were seen in Figure 2. Similar to Lawrence et al. (2018), each "pixel" in Figure 3 represents a seasonal mean difference (in other words, the reanalysis minus REM averaged over a given season) for an individual year and potential temperature level.

Overall, most reanalyses are within $0.3*10^{-4}$ s$^{-1}$of the REM throughout the timeseries. Constant biases with respect the
REM are seen in CFSR/CFSv2, with a bias towards lower magnitudes in each hemisphere of up to about $1*10^{-4}$ s$^{-1}$ around 2500 K throughout the record.

Some other discontinuities are also apparent: For example, ERA-Interim displays a discontinuity near 850 K at the time of TOVS-ATOVS transition in the Southern Hemisphere, and around 1500 K in the Northern Hemisphere. MERRA-2 shows poorer agreement with the REM in the middle stratosphere between 1994 (the discontinuity associated with changes in the
assimilation of the SBUV) and the TOVS-ATOVS transition; this is particularly noticeable in the Southern Hemisphere but also present in the Northern Hemisphere, and is consistent with changes shown by Lawrence et al. (2018) in SH vortex diagnostics. JRA-55 also shows an apparent shift to a high bias in the Northern Hemisphere around 2001 above 2000 K.

## 4.1 Effects of differing calculation methods

PV is a commonly provided product in reanalysis datasets, but mostly on a limited set of isentropic and/or isobaric levels.
As mentioned in section 2, only MERRA-2 provides PV on model levels, while CFSR/CFSv2 provides relative vorticity and ERA-Interim provides absolute vorticity. To help understand how much the method used to calculate PV may affect differences between the reanalyses, Figure 4 shows climatological differences between the sPV based on winds, pressure, and temperature and sPV based on each reanalyses' provided vorticity.

This analysis indicates that the differences arising from different methods for calculating PV are considerably smaller than
the differences found between the REM and the reanalysis fields: The range of the color bar in this figure is 10 times smaller that the one used in Figure 2. For the most part, the differences arising from calculating PV differently are within $0.01*10^{-4}$ s$^{-1}$, and in the worst instances *only* up to $0.1*10^{-4}$ s$^{-1}$ (i.e., the difference from calculating PV in different ways for a single reanalysis is only up to 10% of the difference of that reanalysis' PV from the REM).

Although these differences stemming from different methods of calculating PV are small, such differences can be relevant
to many studies: For instance, the strength of the polar vortex is often assessed using PV gradients along its edge (see Section 7 below), where the largest differences are seen in winter in Figure 4. Further, PV is often used as a part of coincidence criteria, for which small differences might make the difference in comparing an air parcel that was inside the vortex edge with one that was outside, or one that was in the troposphere to one in the stratosphere. In addition, it has previously been shown that many studies are adversely affected by using coarser pressure-level gridded products versus those on native model levels, e.g., for





UTLS studies (e.g., Manney et al., 2017; Tegtmeier et al., 2020). It would thus be valuable if reanalysis centers provided PV on model levels in future reanalysis products, so that it is consistent with all the model physics. For the current reanalyses, it would be useful for users to do their analysis using derived PV on model levels (as opposed to using the provided PV on discrete levels) for situations where those analyses may be sensitive to the exact values of PV or its gradients.

## 5 Equivalent latitude comparison

PV equivalent latitude (EqL in this paper) is a quasi-Lagrangian coordinate defined as the geographical latitude encompassing the same area as given PV contour (Butchart and Remsberg, 1986). EqL is widely used in stratospheric studies, for example, to construct climatologies (e.g. Jones et al., 2012; Koo et al., 2017; Thomason et al., 2018), to compare non-coincident datasets (e.g., Manney et al., 2001; Lumpe et al., 2006; Manney et al., 2007; Velazco et al., 2011), for reanalysis comparisons (e.g., Davis et al., 2017), in UTLS transport analyses (e.g., Haynes and Shuckburgh, 2000; Hegglin et al., 2006; Berthet et al., 2007), and to study polar vortex dynamics and trace gas evolution (Orsolini et al., 2005; Manney et al., 2009; Manney and Lawrence, 2016; Manney et al., 2020, and references therein).

EqL is computed using the $0.5°$ gridded PV fields using a piecewise constant method, where the PV value is assumed constant within each grid cell. Further, EqL is only computed on isentropic surfaces where no more than 5% of data values on that level are missing or bad. This criterion only affects the lowest and highest levels in each reanalysis.

Figure 5 shows the EqL zonal mean REM seasonal climatology (1980 - 2014) as well as its standard deviation. The largest amount of variability is found along the polar vortex edges, as well as at the top of the upper troposphere subtropical jet in all seasons (presumably related to EqL becoming a less appropriate coordinate near / below the tropopause, e.g., Manney et al., 2011; Pan et al., 2012).

Figure 5 also shows EqL differences between the reanalysis fields and the REM with overlaid contours of each reanalyses' EqL climatology. In many regions, the reanalyses and the REM agree within $0.5°$. Pronounced differences, greater than $10°$, are seen near the poles around 2500 K. In contrast to differences in sPV, which are most pronounced during each hemisphere's winter (e.g., Figure 2), the largest EqL differences occur during each hemisphere's summer, when small sPV values make the EqL computation noisier. Similarly, large differences are also found in the tropics (at all levels) where sPV values approach zero.

As with the sPV comparison, to investigate the impact of the discontinuities discussed in section 3, Figure 6 shows EqL yearly differences with respect to the yearly REM. In particular it shows EqL "pixel" differences from the REM for each reanalysis during each hemisphere's summer when the most pronounced EqL differeces were found, as well as, during each hemisphere winter when the EqL REM standard deviation was the largest.

During summer, large biases (greater than $|10°|$) spanning the entire timeseries are seen in CFSR/CFSv2 and JRA-55 near 2500 K. Similar biases are also present in ERA-Interim and MERRA-2 but somewhat mitigated after around 2005. CFSR/CFSv2 shows a bias towards lower magnitudes in each summer hemisphere at most levels, though this becomes smaller in the Southern Hemisphere around or within a few years after the TOVS-ATOVS transition. In contrast, JRA-55 shows a





bias towards higher magnitudes in both hemispheres, which becomes somewhat smaller in the Northern Hemisphere after the TOVS-ATOVS transition. Possible changes in ERA-Interim and MERRA-2 around this transition are not obvious.

During winter, the yearly differences are much smaller, particularly in the Northern Hemisphere where most of the time the differences are within |1°|. In the Southern Hemisphere, the impact of the TOVS-ATOVS is evident for MERRA2, CFSR/CFSv2 and JRA55.

## 6 Dynamical tropopause comparison

Another widespread use of PV is to determine the location of the "dynamical" tropopause. Several PV or PV gradient thresholds have been used in the literature (Holton et al., 1995; Highwood and Hoskins, 1998; Highwood et al., 2000; Stohl, 2003; Schoeberl, 2004; Kunz et al., 2011); we calculate dynamical tropopause characteristics at 2.0, 3.5, and 4.5 potential vorticity units (PVU), which span the range of PV values most commonly used in the literature. Dynamical tropopauses are identified using the JEt and Tropopause Products for Analysis and Characterization (JETPAC) package (e.g., Manney et al., 2011; Manney and Hegglin, 2017). Since PV does not provide a well-defined tropopause in the tropics (e.g., Holton et al., 1995), the dynamical tropopause is defined as the 380 K isentropic surface wherever the PV contour definition would place it at a higher potential temperature. Tropopauses are identified using the reanalysis fields on their full model grid but post-interpolated to 0.5° bins for the comparisons shown here.

Figure 7 shows climatological REM dynamical tropopause altitude maps for different seasons. In particular, it shows the 2PVU climatological dynamical tropopause, which is one of the most commonly used PV-based tropopauses (e.g., Stohl, 2003; Hegglin et al., 2006; Kunz et al., 2011). Overall, this climatology is consistent with previous studies (e.g., Hoinka, 1998; Highwood et al., 2000; Liu et al., 2014; Manney et al., 2014): the tropopause altitude has a strong zonal structure, and a decrease from the equator toward the poles with a sharp latitudinal gradient in both hemispheres around 30° (near the subtropical upper tropospheric jet). The Southern Hemisphere tropopause is, in general, more zonally symmetric than that in the Northern Hemisphere, which is characterized by flow patterns that follow the strong zonal variations of the subtropical jets (e.g., Manney et al., 2014) that are ultimately related to topographic variations and land/sea contrasts.

Figure 7 also shows the differences of each of the reanalyses from the REM, that is, reanalysis − REM, so that positive (negative) values indicate that the reanalysis tropopause altitude is higher (lower) than that of the REM. Generally, the differences are within 0.1 km over most of the globe. Around 30°N and 30°S the difference can be up to 1 km. These relatively large differences are related primarily to small discrepancies in the location of the sharp decrease in tropopause altitude from the tropics to mid-latitudes. Other ∼1 km discrepancies can be found over Greenland and over the Andes mountains and are likely related to differences among the reanalyses parametrizations of orographic gravity waves (e.g., Limpasuvan et al., 2007; Sato et al., 2012), which have a large impact in these regions. Large discrepancies are also seen over Antarctica where conventional data used for reanalysis inputs (e.g., high resolution radiosonde temperature profiles that help constrain the vertical structure) are sparse; thus larger disagreements in this region are expected.





Figure 8 shows yearly zonal mean differences with respect to the yearly REM. In general, there are few obvious improvements over time, though ERA-Interim shows small improvements in the tropics to subtropics after the 1985 discontinuity and after the TOVS-ATOVS transition; its bias decreases from about 0.2 km to about 0.1 km. CFSR/CFSv2 shows what appear to be relatively gradual improvements over time near 30°S in DJF and 30°N and 30°S in JJA; these improvements appear slightly

more significant after the TOVS-ATOVS transition and again more so after the start of assimilation of IASI radiances and the CFSR to CFSv2 transition. Over Antarctica, where the largest difference were seen in Figure 7, there are no clear improvements with time. Thus, the TOVS-ATOVS change in assimilated data that so dramatically affected the Antarctic temperatures (Long et al., 2017; Lawrence et al., 2018) does not appear to have as strong an influence on the PV values demarking the tropopause, perhaps suggesting that either temperature changes are smaller around the tropopause or that those temperature changes do not

strongly impact the static stability profile in the tropopause region; this is consistent with the lack of time changes in reanalysis differences in lapse rate tropopauses found by Xian and Homeyer (2019).

Almost identical conclusions can be drawn for the 3.5 and 4.5 PVU dynamical tropopause altitude, with slightly better agreement in the tropics for these higher PV values.

## 7 Polar vortex comparisons

Most commonly used methods of identifying the vortex edge are based on PV and/or PV gradients (e.g., Nash et al., 1996; Manney et al., 2007; Lawrence and Manney, 2018). In this study we use climatological extreme sPV gradients as a function of equivalent latitude to identify an sPV contour representative of the vortex edge. Following Lawrence et al. (2018), we bin sPV as a function of equivalent latitude, differentiate, and catalog the maximum value between 30° and 80° equivalent latitude. We use data from the extended winter season, that is, from November through April in the Northern Hemisphere and from

April through November in the Southern Hemisphere. Figure 9 shows the sPV of the vortex edge for each reanalysis as well as the mean of these values (i.e., the REM vortex edge sPV). The values, and hence reanalysis differences, are very similar to those used by Lawrence et al. (2018), but not identical because of differences in the numerical methods, e.g., for gradients and interpolations, and the sensitivity of identification of maximum gradients to those calculations. The reanalyses' vortex edge sPV values generally agree quite well in the lower to mid-stratosphere, except for a large maximum sPV values in JRA-55 near

700 K; they diverge slightly more in the upper stratosphere, where the vortex edge starts becoming more difficult to define using PV gradients (e.g., Manney et al., 2007; Harvey et al., 2009), with ERA-Interim tending to be slightly lower and CFSR/CFSv2 slightly higher than the other reanalyses. For the comparisons below, we use the REM vortex edge sPV for all reanalyses.

Figure 10 shows the polar vortices at 850 K for a strong, quiescent vortex, for a vortex displacement sudden stratospheric warming (SSW), and for a vortex split SSW. The purpose of this figure is two-fold: (1) it highlights how complex the polar

vortex is and (2) it shows that despite the overall large-scale similarities among the reanalysis there are some significant differences on individual days, especially in the degree to which the reanalyses PV fields show narrow tongues or small vortex fragments at the sPV of the vortex edge. JRA-55 stands out as not showing such small-scale features in the vortex edge sPV in the vortex displacement and vortex split examples.



To quantify such differences we identify vortices for each day and catalog the number of vortices as well as their area. To identify the vortices on a given isentropic surface, we use a flood filling algorithm – an algorithm that determines pixels meeting a threshold value in a 2-D array – to find all the adjacent pixels that were above/below (depending on the hemisphere) the chosen threshold. Further, we calculate the area of each vortex and filter out those with area smaller that the polar cap area

poleward of 85° (following Lawrence and Manney, 2018). Once the vortices are identified we compute moment diagnostics including equivalent ellipses, centroids, aspect ratios, and angles (measured eastward from 0° longitude) (e.g., Matthewman et al., 2009; Lawrence and Manney, 2018). Figure 10 also shows the REM equivalent ellipses computed for each vortex.

Figure 11 compares the climatological mean vortex area for each reanalysis at several representative levels. This comparison is only for the main vortex (i.e., the biggest one on each day), which in this climatological view encompasses 98% of the total

area occupied by all vortices.

Overall the seasonal variations found in the reanalyses are similar and consistent with seasonal variations found in the literature (e.g., Manney and Zurek, 1993; Waugh and Randel, 1999; Harvey, 2002; Lawrence et al., 2018):

– In both hemispheres, climatological vortex area increases with increasing height. For example, in the Southern Hemisphere the vortex can reach values up to $8*10^7$ km$^2$ at 1300 K (around 30% of the hemisphere) but only up to $4*10^7$ km$^2$

at 440 K (only about 15%). In the Northern Hemisphere the vortex can reach values up to $6.5*10^7$ km$^2$ at 1300 K (22% of the hemispheric area) but only up to $2*10^7$ km$^2$ at 440 K (less than 10% of the area).

– In both hemispheres, the climatological vortex starts forming later at lower altitudes. In the Southern Hemisphere the vortex starts forming around March at 1300 K but not until late April at 440 K. In the Northern Hemisphere, the vortex starts forming around September at 1300 K but around November at 440 K. Thus, there is a lag of about two months

between 1300 K and 440 K for the formation of the vortex.

– In the Southern Hemisphere the polar vortex season finishes earlier at higher altitudes. That is, it breaks down around November at 1300 K but not until January at 440 K.

– In the Northern Hemisphere, the climatological vortex is seen to break down around May at all levels because of much greater variability in degree to which the Arctic vortex breakdown is dynamically-driven, and thus the timing of that

breakdown.

– In the Northern Hemisphere, the vortex area displays a doubly peaked distribution in fall and spring at 1100 and 1300 K.

While the large-scale evolution is similar among the reanalyses, there are some notable differences in the seasonal evolution. Quantitative estimates of these differences are made when all climatological vortices are bigger than $0.15*10^7$km$^2$ (around 0.5% of the hemisphere) to avoid spurious vortices identified during the warm season (particularly at 440 K).

In midwinter, maximum differences vary from about 5% to about 20%, with the largest differences at higher levels (1100 and 1300 K). The most noticeable differences are during the vortex onset and demise. These differences arise from differences in vortex formation and breakup dates and rates, and appear exaggerated at these times when the vortex area is small because





the comparisons are in percent. Figure 12 shows the difference in days between the reanalysis and the REM vortex formation and decay dates. These dates are estimated as the date when the climatological vortex area reaches/diminishes to an area of $0.15*10^7 \text{km}^2$ (around 0.5% of the hemisphere).

The vortex formation and decay dates for the reanalyses are usually within about 4 days of the REM dates for most levels, with many cases where the reanalyses agree to within 2 days (e.g., during demise in the Northern Hemisphere at levels above 600 K). Largest variability is found during onset in the Southern Hemisphere, where the differences can be as large as 9 days at 600 K.

Figure 13 shows a comparison of the formation and decay rates (the change of area per day at the beginning and end of the vortex season). To compute these rates we fitted a line from the start date to the date when the area reached half of its peak value (on its way up), as well as from the date the area reached half of its peak value (on its way down) to its end date. Figure 13-left shows examples of these linear fits. These fits were estimated for the REM and each reanalysis.

Figure 13-right shows a comparison of the slopes of those fits. Overall, below 850 K the reanalyses agree well, within about 10%. At 1100 and 1300 K, the differences can be up to 25%. The largest variability is seen at 1100 and 1300 K in the Northern Hemisphere during demise, consistent with the time and levels that also show most interannual variability and largest differences in vortex area.

Figure 14 shows the shape and position of the climatological monthly mean equivalent ellipses for July and January (other months show similar results) for the southern and Northern Hemisphere, respectively. As shown, the Antarctic vortex is larger, more zonally symmetric, and its shape varies less with height than the Arctic vortex (e.g., Waugh and Randel, 1999). The most variability among the equivalent ellipses is seen at 1100 K and 1300 K, consistent with the variability in area seasonality (up to 20% in midwinter). Most of the reanalyses agree remarkably well at lower levels, with the exception of CFSR/CFSv2 at 440 K, which shows a clear departure from the other reanalyses in the Northern Hemisphere. MERRA-2 also shows slightly smaller ellipses than the other reanalyses at 1300 K.

To investigate the impact of the discontinuities discussed in section 3, Figure 15 shows the vortex area differences between each reanalysis and the REM as function of potential temperature and year. Some clear discontinuities in the vortex area agreement can be seen. In the Southern Hemisphere, ERA-Interim shows a discontinuity around 2002 and MERRA-2 shows a discontinuity after 2004, showing better agreement with the REM in the later years. The ERA-Interim discontinuity is not obviously related to any of the transitions discussed in section 3. JRA-55 displays a discontinuity at the TOVS-ATOVS transition in both hemispheres, with better agreement during the ATOVS period in the Southern Hemisphere, and changes in the vertical distribution of biases in the Northern Hemisphere. CFSR/CFSv2 consistently overestimates the REM area by more than 5% around 1200 K throughout the record.

Figure 15 also shows the equivalent ellipses' aspect ratio differences as a function of potential temperature and year. Discontinuities can be seen in MERRA-2 and in JRA-55 in the Northern Hemisphere around the TOVS-ATOVS transition with MERRA-2 showing better agreement with the REM around 1200 K and with JRA-55 showing worse agreement around 1000 K during the ATOVS period. The most noticeable difference is found in the Northern Hemisphere in the CFSR/CFSv2 compari-





son, with an underestimation with respect to the REM of up to 10% of the aspect ratio around 1200 K throughout most of the record.

Lastly, Figure 15 also shows the equivalent ellipses angle differences, that is, the difference in the orientation of the equivalent ellipses, as a function of potential temperature and year. In the Southern Hemisphere, the four reanalyses agree with the REM
overall within 1° throughout the entire timeseries with no discernible patterns or discontinuities. The Northern Hemisphere shows more variability in the agreement among the reanalysis and the REM but still agrees overall within about 3°. This relatively larger variability is expected because of the larger variability and asymmetry in the Northern Hemisphere vortex. No clear discontinuities are seen.

## 8   Summary

In this study we intercompare PV from several reanalyses (ERA-Interim, MERRA-2, CFSR/CFSv2, and JRA-55) using the REM as a reference. This intercomparison is performed not only directly on the PV fields but also on commonly used PV-derived metrics such as equivalent latitude, the location of the dynamical tropopause, and stratospheric polar vortex characteristics. Since MERRA-2 is the only reanalysis that provides PV fields on model levels, we derived PV for the rest of the reanalysis using their available model level products.

The main findings can be summarized as follows:

- In the zonal mean, sPV from all reanalyses agrees with the REM within $0.1*10^{-4}$ s$^{-1}$. The most pronounced differences are for CFSR/CFSv2 during each hemisphere's winter (and to a lesser degree during fall), with a low bias of up to $1*10^{-4}$ s$^{-1}$ (i.e., up to 40%).

- Differences related to using different PV calculation methodologies (since only MERRA-2 provides PV on model levels)
are usually within $0.01*10^{-4}$ s$^{-1}$, and in the worst instances *only* up to $0.1*10^{-4}$ s$^{-1}$ (around 10% of the difference of the reanalyses' PV from the REM). Although these differences are small, we recommend that reanalysis centers provide PV on model levels in future reanalysis products so that it is consistent with all the model physics. For the current reanalyses, we recommend that users derive PV from the reanalyses' provided vorticity (or winds and temperature) on model levels for any calculations that may be sensitive to exact values (including "threshold" calculations such as
the dynamical tropopause and stratospheric vortex edge) as opposed to (or in addition to) using the provided PV on coarsely-resolved sets of discrete levels.

- Pronounced EqL differences, greater than 10°, occur during each hemisphere's summer, when sPV values are small (which makes the EqL computation noisier), particularly near 2500 K in the polar regions. Differences greater than 10° are also found in the tropics where sPV values are also small.

- Climatological dynamical tropopause differences are within 0.1 km over most of the globe. Larger discrepancies were found around 30°N and 30°S (near the location of the subtropical upper tropospheric jet and the "tropopause break") where mismatches in the location of the sharp decrease in tropopause altitude from the tropics to mid-latitudes are



so common as to affect the climatology; over Greenland and the Andes regions affected by differences among the reanalyses' parametrizations of orographic gravity waves; and over Antarctica, where conventional input data are most sparse.

– The most noticeable polar vortex area differences are during the vortex formation and demise; these arise primarily because of slightly different vortex season start and end dates, and different formation and decay rates. In midwinter the maximum differences vary from about 5% at lower levels (440–850 K) to about 20% at higher levels (1100 and 1300 K).

– Largest variability among the reanalyses' polar vortex equivalent ellipses is found at 1100 and 1300 K, consistent with the variability and reanalysis differences in area seasonality. At lower levels, most of the reanalyses agree remarkably well, except for CFSR/CFSv2 at 440 K, which shows a clear departure from the other reanalysis in the Northern Hemisphere.

– The discontinuities associated with changes in assimilated data or changes in processing streams that drastically affect other reanalysis fields (such as temperature and temperature-based diagnostics) (e.g., Long et al., 2017; Lawrence et al., 2018) do not appear to typically have a strong impact on sPV, EqL, dynamical tropopause altitudes, or polar vortex characteristics based on PV.

The overall good agreement of PV among the reanalyses provides some confidence that any of these recent reanalyses are
15 appropriate for most studies using PV-based diagnostics. The areas where we find some disagreement do, however, suggest that caution should be used, and that comparing multiple reanalyses is important for studies that show strong sensitivity to exact PV values.

*Data availability.* The datasets used are publicly available, as follows:

– MERRA-2: https://disc.sci.gsfc.nasa.gov/uui/datasets?keywords=%22MERRA-2%22
– ERA-Interim: http://apps.ecmwf.int/datasets/
– JRA-55: Through NCAR RDA at http://dx.doi.org/10.5065/D6HH6H41
– CFSR/CFSv2 model level data: Available upon request from Karen H Rosenlof (karen.h.rosenlof@noaa.gov)
– JETPAC tropopause products: Contact Gloria L Manney (manney@nwra.com) or Luis F Millán (lmillan@jpl.nasa.gov)

*Author contributions.* LFMV and GLM designed the study. LFMV wrote most of the algorithms used and carried out the analyses. ZDL
provided the climatological extreme sPV gradients used to identify the polar vortex and provided the equivalent ellipses algorithm. GLM provided scientific expertise throughout all stages of the research. LFMV wrote the paper, and GLM and ZDL commented on and edited the manuscript.

*Competing interests.* There are no competing interests.



*Acknowledgements.* LMV's research was carried out at the Jet Propulsion Laboratory, California Institute of Technology, under a contract with the National Aeronautics and Space Administration. GLM was supported by a subcontract from the JPL Microwave Limb Sounder project.





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



**Figure 1.** (top panel) sPV REM timeseries. (bottom panels) sPV anomalies with respect to each reanalyses' own monthly climatological values. Vertical lines show the discontinuities described in the text.

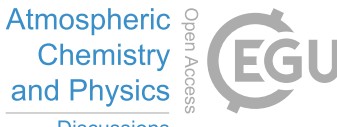
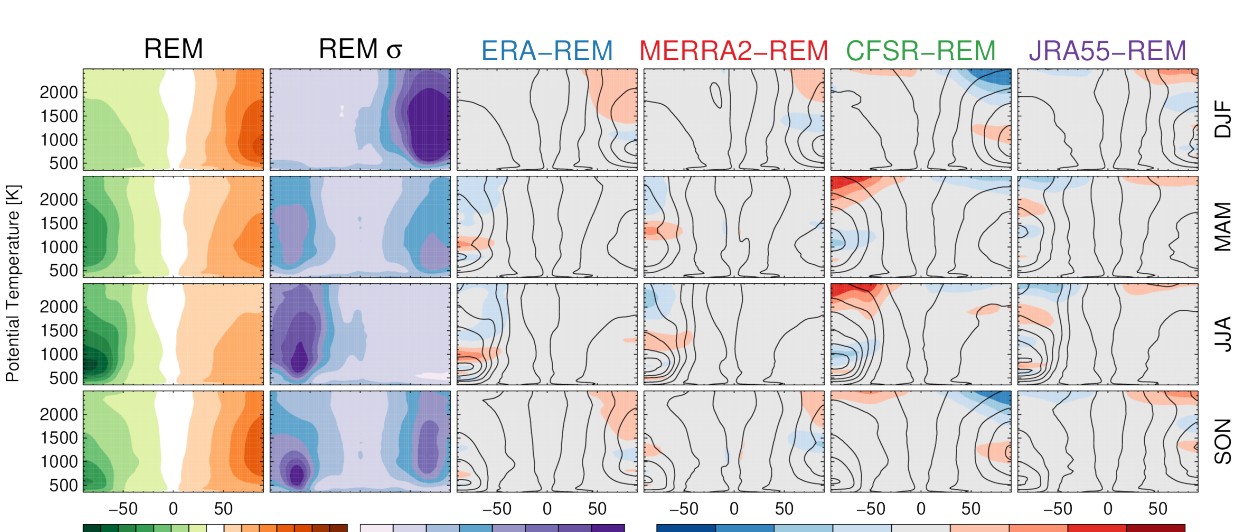

**Figure 2.** (left columns) sPV REM seasonal climatological mean and standard deviation. (right columns) sPV difference between each reanalysis and the REM, line contours show each reanalyses' seasonal climatology.

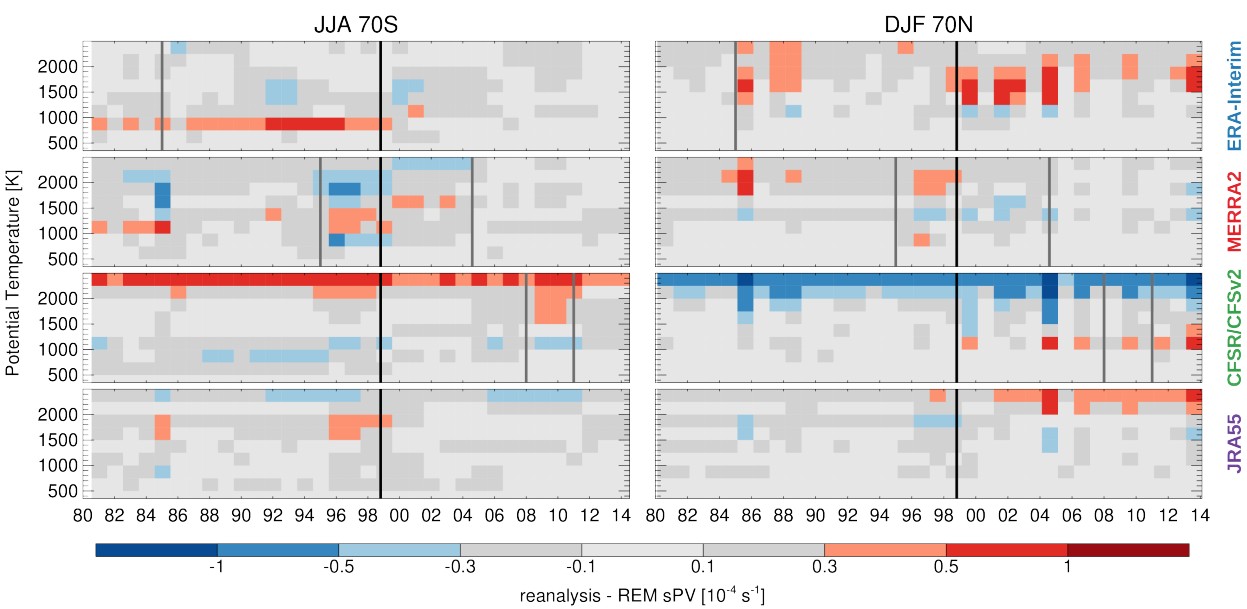

**Figure 3.** sPV differences for each reanalysis from the REM as a function of year and potential temperature. (left) JJA at 70°S and (right) DJF at 70°N differences. Vertical lines show the discontinuities associated with changes in assimilated datasets or different processing streams.



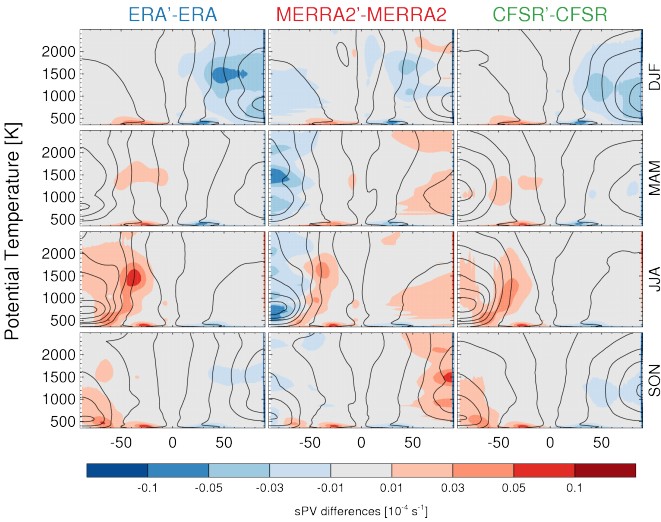

**Figure 4.** Differences between the sPV from each reanalysis provided vorticity or PV and the sPV computed from that reanalyses' horizontal wind, pressure, and temperature fields. Overlaid contours show each reanalyses' climatology based on that reanalyses' provided vorticity.

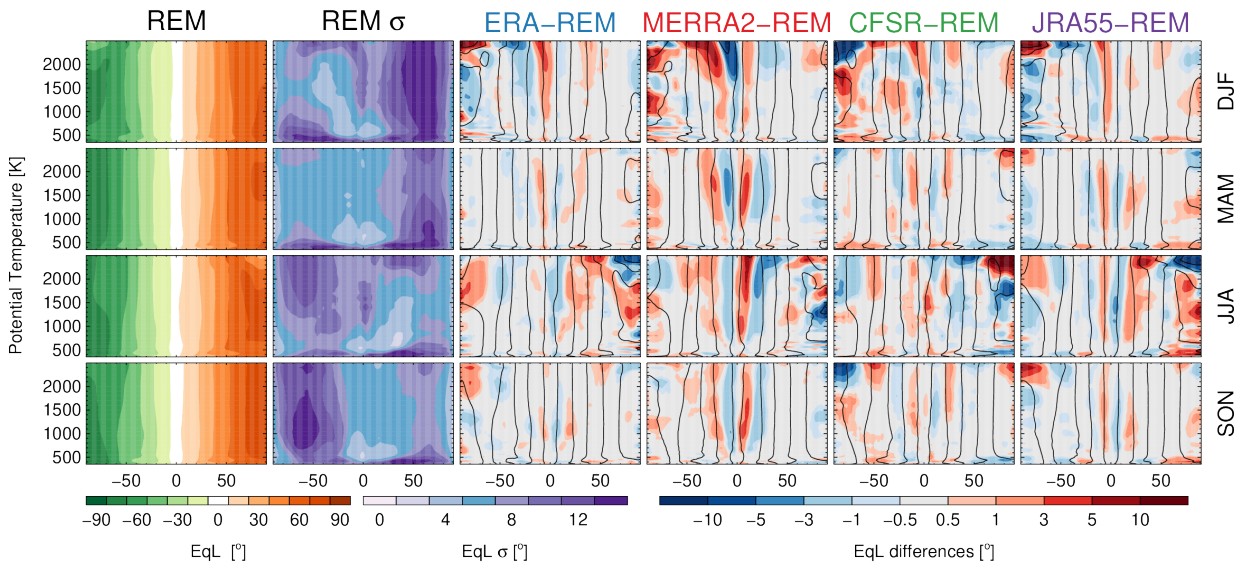

**Figure 5.** (left columns) Equivalent latitude REM seasonal climatological mean and standard deviation for different seasons. (right columns) Equivalent latitude difference between each reanalysis and the REM. Overlaid contours show each reanalysis fields respective climatology.



**Figure 6.** EqL differences for each reanalysis from the REM as a function of year and potential temperature. (top panels) summer hemispheric differences, (bottom panels) winter hemispheric differences at 70°S and 70°N. Vertical lines show the discontinuities associated with changes in assimilated datasets or different processing streams.





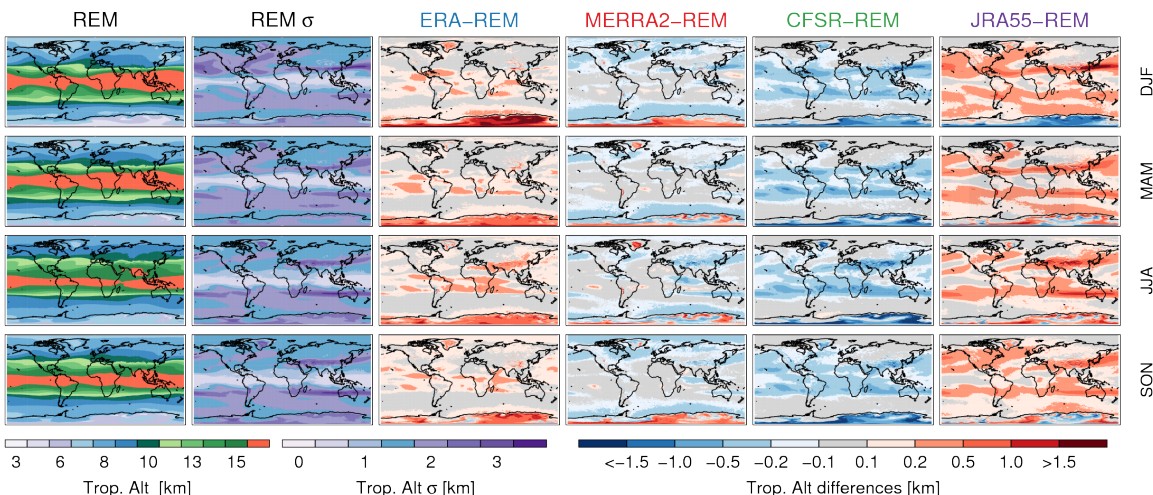

**Figure 7.** (left columns) Climatological REM 2 PVU tropopause altitudes and standard deviations for different seasons. (right columns) Difference of each reanalysis from the REM.



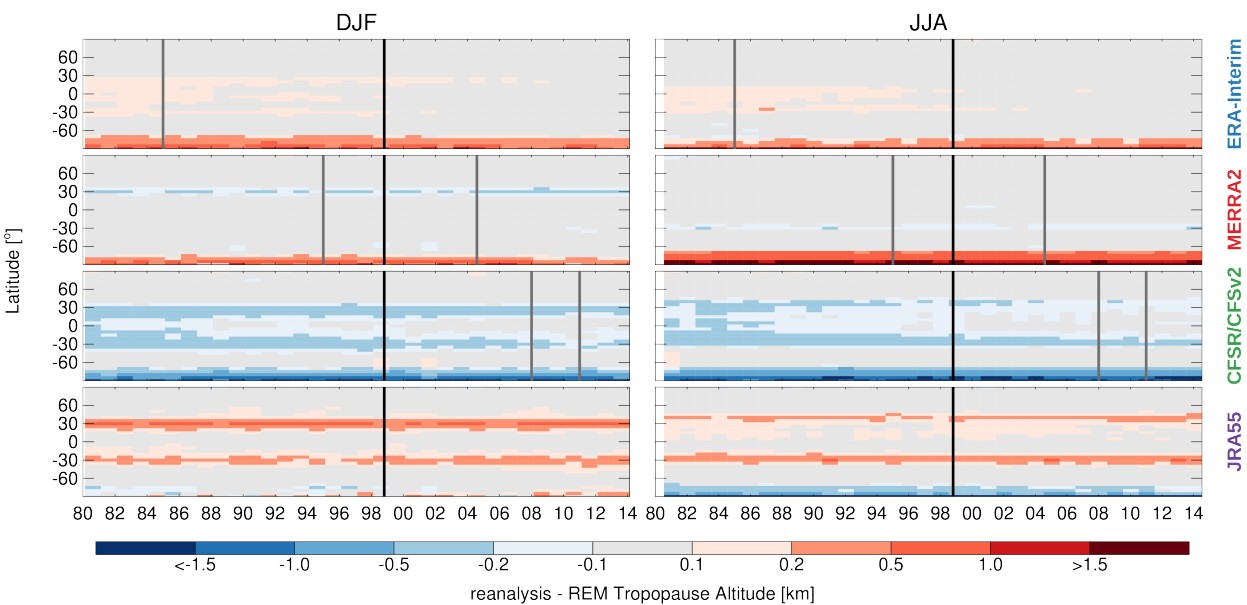

**Figure 8.** 2 PVU tropopause altitude zonal mean differences for each reanalysis from the REM as a function of year. (left) DJF and (right) JJA differences. Vertical lines show the discontinuities associated with changes in assimilated datasets or different processing streams.

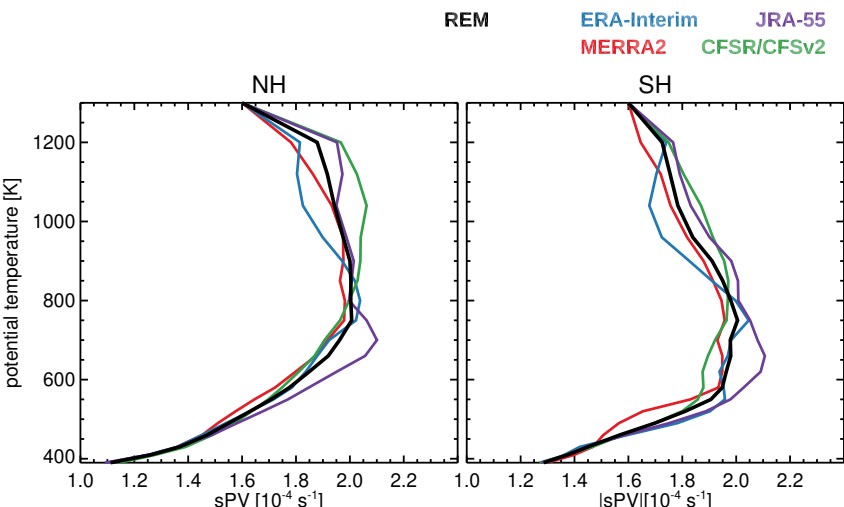

**Figure 9.** sPV vortex definition for each reanalysis as well as for the REM. These were constructed using climatological values from 1980–2014 data.





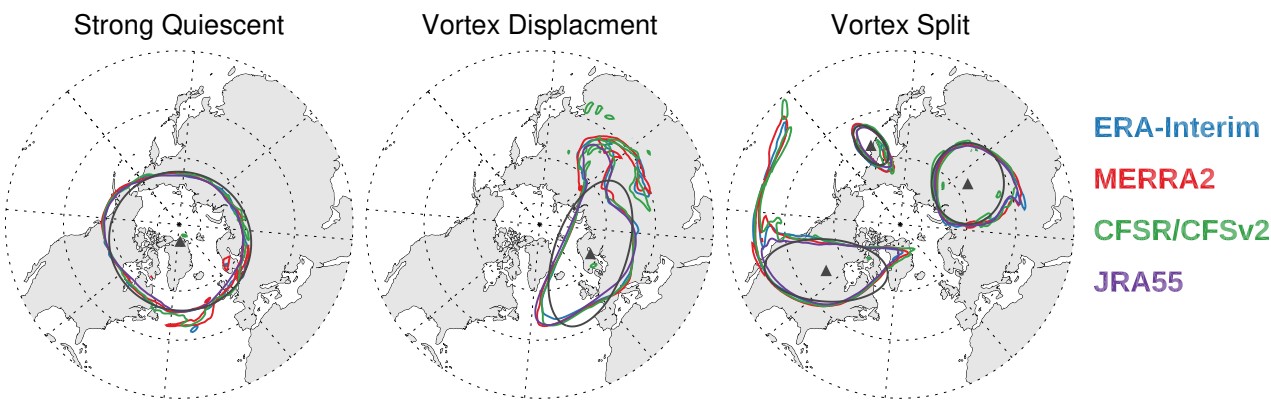

**Figure 10.** sPV maps at 850 K during a strong quiescent period of the polar vortex, during a vortex displacement, and during a vortex split. These maps are for 8 January 2009, 23 January 1987, and 25 January 25 2009, respectively. These maps show the raw sPV fields, i.e., vortices were not excluded by the minimum area threshold. Dark gray lines show equivalent ellipses computed for the REM sPV for these days, triangles represent the center of those equivalent ellipses.



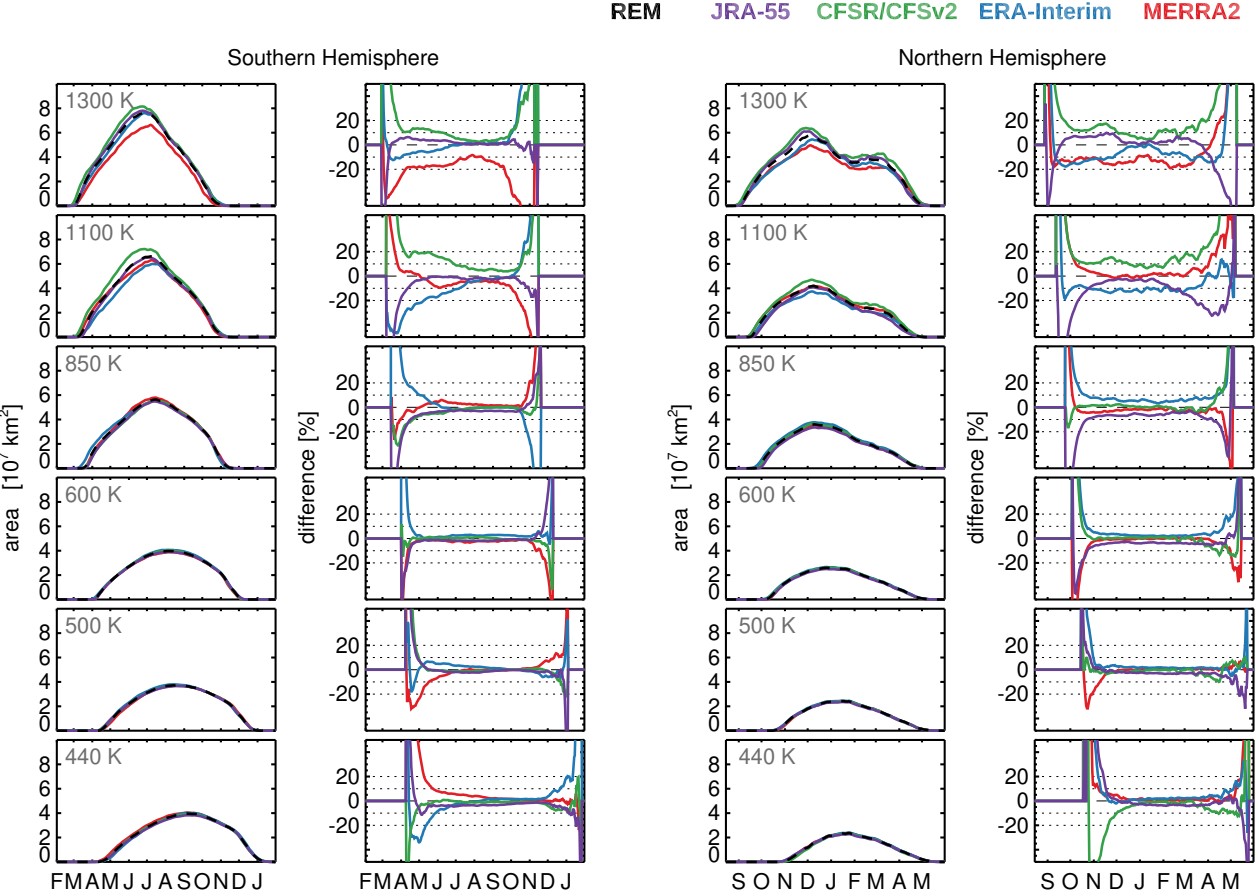

**Figure 11.** Primary (i.e., largest) polar vortex area seasonal evolution during 2005–2014 in the Southern (left two columns) and Northern (right two columns) hemisphere. Right column for each hemisphere shows differences with respect to the REM; as a guideline the dashed gray lines show ±10% and ±20% differences.





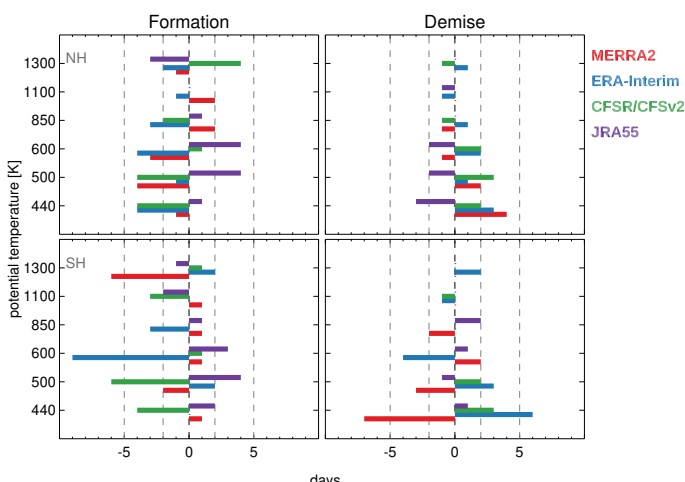

**Figure 12.** Climatological difference in days between the reanalysis and the REM vortex formation and decay dates.



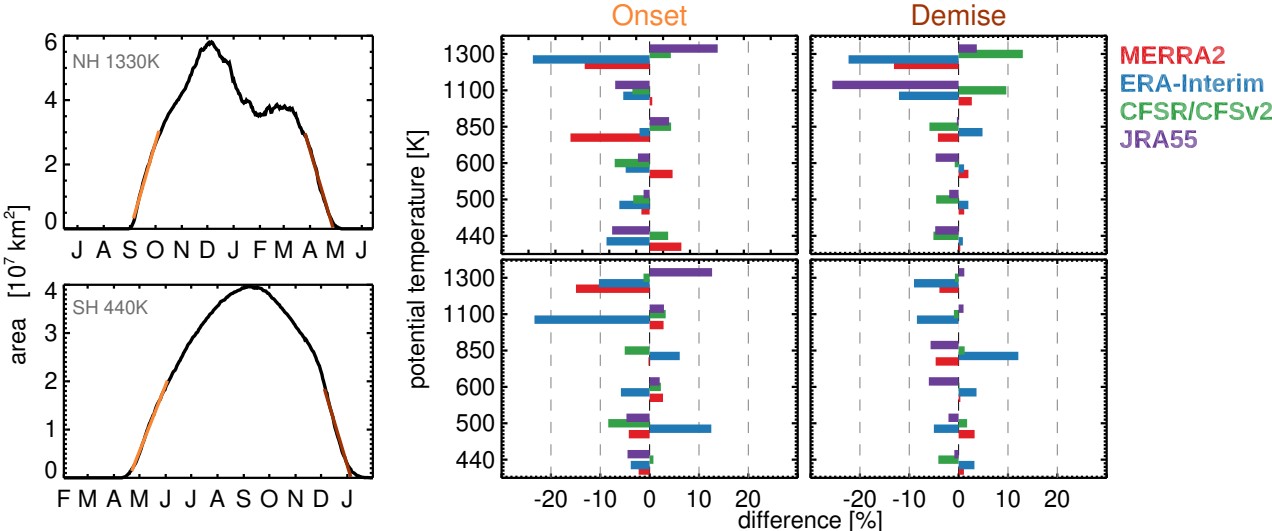

**Figure 13.** (left) Examples of the linear fits applied during onset and demise of the vortex in the REM (see text). (right) Percent difference between the linear fits' slopes for each reanalysis from the REM; (top) Northern Hemisphere and (bottom) Southern Hemisphere.





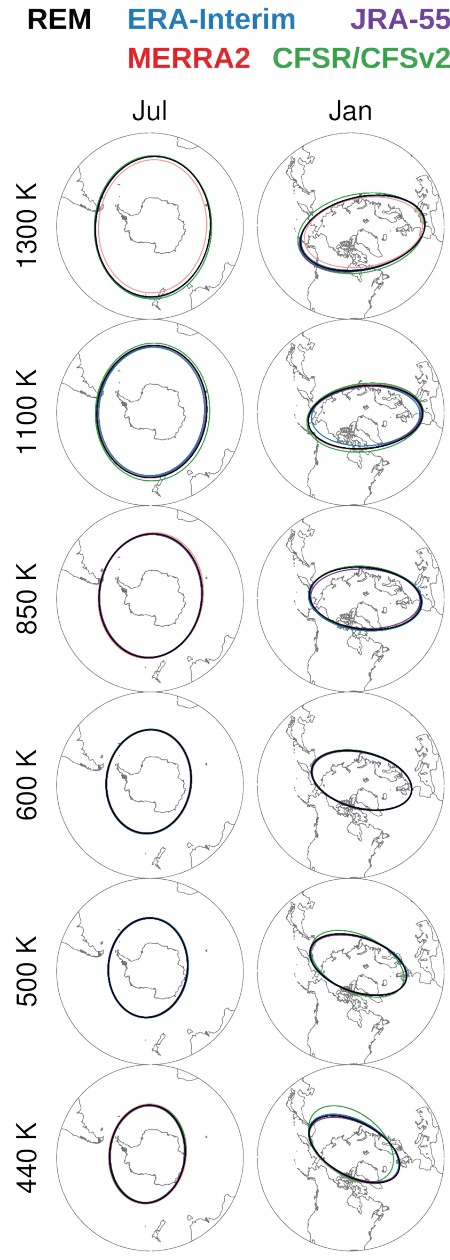

**Figure 14.** Polar stereographic plots of climatological monthly-mean equivalent ellipses from each reanalysis at several potential temperatures. (right) July monthly mean equivalent ellipses for the Southern Hemisphere; (left) January equivalent ellipses for the Northern Hemisphere





**Figure 15.** Vortex area, aspect ratio, and angle differences for each reanalysis from the REM as a function of year and potential temperature. Only vortices greater than $0.15*10^7 \text{km}^2$ were considered. Vertical lines show the discontinuities associated with changes in assimilated datasets or different processing streams.



**Table 1.** Basic specifications of the reanalysis forecast models.

| Reanalyses | Grid | # levels | Lid Height | Main Reference |
|---|---|---|---|---|
| MERRA-2 | $0.625°$x$0.5°$ | 72 | 0.01 hPa | Bosilovich et al. (2015) |
| ERA-Interim | $0.75°$x$0.75°$ | 60 | 0.1 hPa | Dee et al. (2011) |
| CFSR/CFSv2 | $0.5°$x$0.5°$ | 64 | $\sim$0.26 hPa | Saha et al. (2010) |
| JRA-55 | $0.56°$x$0.56°[a]$ | 60 | 0.1 hPa | Kobayashi et al. (2015) |

[a] approximately, these fields are provided on a Gaussian grid.