# Peer review of "Reanalysis intercomparison of potential vorticity and potential-vorticity-based diagnostics"

_Atmospheric Chemistry and Physics, 2020_

## Referee Comment (RC1) · Anonymous Referee #1 · 4 Jan 2021

This study, as part of the S-RIP, investigates the agreement of potential vorticity diagnostics among four modern reanalysis datasets. Raw PV, PV-based tropopause height, and PV-based polar vortex shape diagnostics are evaluated. The general conclusion is that we can have confidence in using any of these datasets for most studies of the stratosphere using potential vorticity. Many of the diagnostics presented in this work were demonstrated to be useful in previous literature and are, to my knowledge, assessed and compared among a comprehensive set of modern reanalysis datasets for the first time. This comparison will serve as a useful reference for any study investigating stratospheric physics with the use of PV. I thus believe that it can constitute a valuable contribution to the ACP's S-RIP special issue after some rather minor changes.

[Figure]

General comments:

In the discussion associated with Fig. 2, the authors indicate how large the biases are with respect to the climatological PV values. I believe it would be useful to also discuss how large these biases are with respect to interannual or intraseasonal PV variability. Such diagnostics would be especially useful for those interested in dynamical variability on short time scales such as SSW events. Along the same line of thinking, it would be useful to show the root mean square of the bias (calculated from daily values) to capture biases associated with interannual and intraseasonal variability (which may cancel out when averaged over a long period and give an apparent high skill).

Equivalent latitude: It is an important diagnostic evaluated in this paper but is not described in much detail. It could be useful to add an equation describing the relationship between a specific PV contour and its equivalent latitude. Also, what is the reference PV value of the equivalent latitudes reported, the zonal mean PV?

Minor comments:

P5 L26 That the -> than the

P6 L28 differences

P9 L4 That the -> than the

P5 L 4 Could you indicate here that the chosen thresholds are taken from Fig. 9.

P9 L11 These seasonal variations found in the literature, are they found in reanalyses too, or observations?

P11 L22 It is recommended that reanalysis centers provide PV on model levels for greater consistency with model physics. Should it be calculated before or after the reanalysis increment? If the latter, is it really more consistent with model physics?

---

## Referee Comment (RC2) · Anonymous Referee #2 · 12 Jan 2021

Review of "Reanalysis intercomparison of potential vorticity and potential-vorticity-based diagnostics" by Millán et al.

Millán and colleagues study the differences of potential vorticity (PV) and PV-based diagnostics in four modern reanalyses (ERA-Interim, MERRA-2, JRA-55 and CFSR/CFSv2). The discussion centers around (i) the calculation of PV and the differences arising in this task in the various reanalyses, (ii) the impact of the data assimilation on PV in each reanalysis product, (iii) seasonal and annual mean variability of sPV between the various reanalyses, as well as of PV-based diagnostics such as (iv) equivalent latitude, (v) dynamic tropopause and (vi) polar vortex characterization. The major finding is that PV agrees well between the various data sets on the time scales studied in this work. The authors also highlight the situation where more caution is necessary when working with PV. Some differences between the various data sets arise in particular for (i) equivalent latitude calculations at low latitudes or high altitudes, (ii) the dynamic tropopause in regions of jetstreams and of strong topography, as well as (iii) during the formation and demise of the polar vortex.

This work is intended as part of the S-RIP special issue where it perfectly fits. Such a comparison of PV from different reanalysis data sets has not been completed yet, although PV from reanalysis is a widely used diagnostic to analyze transport and dynamics in the troposphere and stratosphere. The questions asked in the paper are clear. The analysis is very convincing; data and methods are well described. The figures are well structured and clear to understand. The conclusions are based on the analysis. It is easy to follow the thoughts of the authors and as a reader I have the feeling that the authors really know what they are talking about. In general I think this study will be of great value to users of reanalysis data and as such I would support publication of this study in ACP in the S-RIP special issue. I have some rather minor comments listed below, which the authors might consider for a revised version.

**Comments:**

- P3, L11: To my knowledge, ERA-I provides relative vorticity, see eg. the ERA-Interim data catalogue: https://apps.ecmwf.int/archive-catalogue/? stream=oper&levtype=ml&expver=1&month=jan&year=1979&type=an&class=ei
- Equation 1: maybe it is worth mentioning that this is the synoptic approximation of  $\ensuremath{\mathsf{PV}}$
- Equation 2: I think, since not everybody might be familiar with sPV, some readers would benefit from a comparison of sPV and PV, maybe shown here for one reanalysis but for different averaging times, e.g., a snapshot, monthly, or yearly mean.
- P3, I26: Could you mention the potential temperature range here.
- P4, I34: Could you say something about the cause of the low bias of sPV? Is this an effect of vertical grid spacing or model physics (e.g. GW drag)?
- Sec. 4: When there is a 4.1, there should be a 4.2 as well. I would suggest to find a subhead for the first paragraphs of this section.
- Figure 4 and related discussion: Is Fig. 4 based on the temporal and zonal mean of the entire data set? Can you say something about the order of the magnitude of the differences, if shorter time periods are considered (monthly means or even shorter).

- P6, L8: climatologies "of what" ?
- P6, L13-15: Is there a reference for the EqL computation used here, so that a reader may be able to look up the computation in detail?
- P6, L17: This is potentially the only greater question which I have. Generally, when you speak of variability, here it is talked about the variability along the polar vortex edge, is this a variability caused by the fact that the reanalyses differ in the representation of the atmospheric features among each other, or because there is a large natural variability of the feature. Maybe it could help to also look at the variability of the shown quantities in individual reanalysis data sets to show whether these already have a large variability or not.
- P7, L7: Do you search the tropopause from top or bottom or asked differently do you refer here to the lowest or highest tropopause in the presence of multiple tropopause as can occur in the vicinity of tropopause folds?
- P7, L29: I wonder whether the parameterizations of orographic GW really have an effect on the tropopause altitude or whether the effect seen here is rather related to larger, resolved GWs themselves above such orography. As far as I know the standard orographic GW parameterizations rather affect higher altitudes by dumping energy at a specified level somewhere in the middle to upper stratosphere and thus affecting the resolved mean flow at those altitudes but not at the tropopause level. Could this here be also a result of the data assimilation, since these are regions with relatively frequent GW occurrences which might be included in radiosonde data which become assimilated?
- In the discussion of Fig. 7 starting on P7, I24, I wonder how much the shown differences could be related to the vertical grid spacing of the individual reanalyses? These data sets all differ in their absolute vertical grid spacing as well as the interpolation may be dependent on the actual location of the individual model levels in the tropopause region. Maybe it would be worth adding information about the vertical grid spacing of the reanalysis products in the tropopause region/stratosphere.
- P9, I4: ...smaller THAN the polar cap
- affiliation 2 and 6 are the same

---

## Referee Comment (RC3) · Katherine Emma Knowland (Referee) · 22 Jan 2021

Review of "Reanalysis intercomparison of potential vorticity and potential-vorticity-based diagnostics" by Luis Millan et al. submitted to Atmospheric Chemistry and Physics.

Summary: In this manuscript, the authors present a thorough intercomparison of reanalysis potential vorticity and diagnostics related to potential vorticity. One modern reanalysis product from four of the major global operational and research centers are selected: CFSR/CFSv2, ERA-Interim, JRA-55 and MERRA2. As part of the SPARC Reanalysis Intercomparison Project, this paper provides clear advice to users of reanalysis PV diagnostics as well as recommends reanalysis centers to consider including PV on model levels in future products for optimal comparisons and scientific studies in the future. My one concern is the number (15) and size (multi-panel) of the figures relative to the text, however I do not see any that can be cut down in size or removed from the main text and included in a supplemental instead. Therefore, I recommend this paper for publication after my minor and technical comments below are addressed.

Comments:

Pg 1 Line 8, Pg 2 Line 32: Add "NASA" before "Modern" since the reanalysis center for the other three products is given.

Pg 2 Line 21: I think the comma after "Nash et al., 1996)," should either be a semi colon or a period.

Pg 3 Line 4: Update Table 1 MERRA2 reference to match Gelaro et al. 2017 reference used here.

Pg 3 Line 4:  Is there a reason ERA-Interim is used instead of ERA-5, the latest ECWMF reanalysis?

Pg 3 Line 7: Why is the period 1980 through 2014 used?

Pg 3 Line 11, Pg 5 Line 21: I checked the ERA-Interim website and I see "Vorticity (relative)" for ERA-Interim on Model Levels. https://apps.ecmwf.int/datasets/data/interim-full-daily/levtype=ml/

Pg 3 Line 26:  ERA-Interim and JRA-55 have coarser resolution than 0.5x0.5 degree. Could this impact your study?

Pg 4 Line 9, Pg 5 Line 2:  Correct me, but I do not see the different processing streams discussed or labelled on Figure 1 or 3, so do we assume that this does not impact PV anomalies? There is mention of a "CFSR to CFSv2 transition" on Pg 8 Line 6 but when this occurred is not stated.

Pg 4 Line 12: Should "regions" be singular since only the south pole is referenced here and not both poles?

Pg 4 Line 32:  Can the authors comment on the fact that the CFSR differences are of opposite sign to the other reanalyses.

Pg 5 Line 11: Can the authors comment on what may cause this difference at CFSR at 2500K?  Is this at all related to the model resolution or how the model treats the upper atmospheric levels? or that CFSR has the lowest Lid height (0.26 hPa, Table 1)?

Pg 5 Line 12: "near 850K" looks more like 850-1000K to me. Are the pixels centered on an isentropic level?

Pg 6 Line 1: UTLS is not defined.

Pg 6 Line 15: The lowest levels are not of interest to this study but can the authors comment on the "highest levels". Does the top of the atmosphere change between the models that this criterion matters at all?

Pg 6 Line 17: It is hard to see along the bottom of the figure. What is the lower limit on Figure 5? Is it 400 K?

Pg 6 Line 21: Suggest adding "; however" connecting these two sentences.

Pg 6 Line 31-Pg 7 line 2: Can the authors comment on why this might be?

Pg 7 Line 4: This looks less evident for MERRA2 after 2005

Pg 7 Line 19: is altitude above sea level or above surface (ground-level)?

Pg 7 Line 26: "can be up to 1 km", does this have anything to do with the difference model resolution in the UTLS?

Pg 8 Line 12: add "(not shown)"

Pg 8 Line 14: In general for this section, do the authors use the native resolution or the 0.5x0.5 degree interpolated resolution?

Pg 9 Lines 30-31: I suggest referencing Figure 11 here since later in this paragraph you reference Figure 12.

Pg 10 Line 3: Is there a reference for selecting this vortex area of $0.15*10^7$ km^2?

Pg 10 Line 17: capitalize "southern"

Pg 10 Line 20: Looks to me CFSR is at 500K and 440K.

Pg 10 Line 22: Suggest moving the sentence starting with "MERRA-2" to after "midwinter)." to keep the upper-level discussion together.

Pg 10 Line 26: add "both" before "showing"

Pg 10 Line 26-27: Could the discontinuity in ERA-I be related to a change in processing streams?

Figures:

Figure 3: The y-axis has minor ticks which seem to be greater than the resolution of the pixels. I recommend reducing the minor ticks. Can annual minor ticks be added to the x-axis on this and the other figures?

Figure 7: Can the y-axis include 30degree latitude interval labels since it is referenced several times on Page 7.

---

## Author Response (AR1)

We thank the reviewers for their comments.

Below are our responses in blue. The biggest change is that the update version includes new figures in the appendix showing root mean square (RMS) differences of the parameters studied to get an idea of the day-to-day variability. Brief text explaining these RMS differences was added throughout the manuscript.

**Reviewer 1**

This study, as part of the S-RIP, investigates the agreement of potential vorticity diagnostics among four modern reanalysis datasets. Raw PV, PV-based tropopause height, and PV-based polar vortex shape diagnostics are evaluated. The general conclusion is that we can have confidence in using any of these datasets for most studies of the stratosphere using potential vorticity. Many of the diagnostics presented in this work were demonstrated to be useful in previous literature and are, to my knowledge, assessed and compared among a comprehensive set of modern reanalysis datasets for the first time. This comparison will serve as a useful reference for any study investigating stratospheric physics with the use of PV. I thus believe that it can constitute a valuable contribution to the ACP's S-RIP special issue after some rather minor changes.

General comments:

In the discussion associated with Fig. 2, the authors indicate how large the biases are with respect to the climatological PV values. I believe it would be useful to also discuss how large these biases are with respect to interannual or intraseasonal PV variability.

Such diagnostics would be especially useful for those interested in dynamical variability on short time scales such as SSW events. Along the same line of thinking, it would be useful to show the root mean square of the bias (calculated from daily values) to capture biases associated with interannual and intraseasonal variability (which may cancel out when averaged over a long period and give an apparent high skill).

We will add the following figures in an appendix:

[Figure]

Caption: Seasonal root-mean-square (RMS) daily (1980-2014) sPV differences.

[Figure]

Caption: Seasonal root-mean-square (RMS) daily (1980-2014) EqL differences.

[Figure]

Caption: Seasonal root-mean-square (RMS) daily (1980-2014) 2PVU dynamical tropopause altitude differences.

[Figure]

Caption: Root-mean-square (RMS) daily (1980-2014) differences. (left) RMS vortex area difference, (middle) RMS aspect ratio difference, (right) RMS equivalent ellipse angle difference.

We will add the following text:

In page 5 line 3: "Root mean square (RMS) daily sPV differences (see Figure A1) show agreement better than 0.3*10−4s−1 throughout most of the atmosphere. RMS differences up to 1*10−4s−1 can be found near the poles in the regions of high sPV variability as shown in Figure 3. These RMS differences capture biases that could be encountered in day by day comparisons that may be important for studies using short time scales such as analysis of sudden stratospheric warming (SSW) events."

In page 7 line 6: "RMS daily EqL differences (see Figure A2) vary from 3 to 10º throughout most of the levels. "

In page 8 line 14: "RMS daily tropopause altitude differences (see Figure A3) are up to 1 km over most of the globe, and greater than 2 km around 30N and 30S, over Greenland and the Andes, and over Antarctica."

In page 11 line 21: "RMS daily vortex area differences (see Figure A4) can be up to 20% in the Southern hemisphere and vary from 20% to 60% in the Northern hemisphere, with the largest differences at around 1200 K. The exceptions are the RMS differences for CFSR/CFSv2 which can differ up to 80% from the REM at this level. "

In page 11 line 25: "RMS daily aspect ratio differences (see Figure A4) are around 10 to 15% in the southern hemisphere and vary from 10% to 40% in the northern hemisphere, with the largest differences around 400-600 K. "

In page 11 line 32: "RMS daily angle differences (see Figure A4) can be up to 50º in both hemispheres, with the exception of CFSR/CFSv2, which can be up to 70º around 440 K, consistent with the orientation departure shown in Figure 15."

In the summary, page 13 line 5 we added: "Day to day variations among the reanalysis (quantified through the RMS differences) suggest that caution should be used when using daily fields and that using multiple reanalyses in such studies is desirable."

Equivalent latitude: It is an important diagnostic evaluated in this paper but is not described in much detail. It could be useful to add an equation describing the relationship between a specific PV contour and its equivalent latitude.

We will add:
**"EqL is computed as,**
**Eql = sin-1 (A/2piR$^2$ -1)**
**where A= A(q) is the area in which PV is less than q on a particular isentropic surface, and R is the radius of the Earth.**
EqL is computed using the 0.5 gridded PV fields using a piecewise constant method, where the PV value is assumed constant within each grid cell. **Simply, for each PV value, on a given isentropic surface, we sum the areas for all grid cell with smaller field values**. Further, EqL is only …."

Also, what is the reference PV value of the equivalent latitudes reported, the zonal mean PV?
That is correct.

Minor comments:

P5 L26 That the -> than the   Done

P6 L28 differences Done

P9 L4 That the -> than the Done

P5 L 4 Could you indicate here that the chosen thresholds are taken from Fig. 9.   We added in brackets, "as shown in Figure 9. "

P9 L11 These seasonal variations found in the literature, are they found in reanalyses too, or observations? In analysis / reanalysis, we will change the sentence too:  "Overall the seasonal variations found in the reanalyses are similar and consistent with seasonal variations found in previous analysis / reanalysis …"

P11 L22 It is recommended that reanalysis centers provide PV on model levels for greater consistency with model physics. Should it be calculated before or after the reanalysis increment? If the latter, is it really more consistent with model physics?

It should be after the increment so that it is consistent with (T, q, U, V, etc), we will change the sentence to: "Although these differences are usually small, we recommend that reanalysis centers provide PV on model levels in future reanalysis products."

**Reviewer 2**

Millán and colleagues study the differences of potential vorticity (PV) and PV-based diagnostics in four modern reanalyses (ERA-Interim, MERRA-2, JRA-55 and CFSR/CFSv2). The discussion centers around (i) the calculation of PV and the differences arising in this task in the various reanalyses, (ii) the impact of the data assimilation on PV in each reanalysis product, (iii) seasonal and annual mean variability of sPV between the various reanalyses, as well as of PV-based diagnostics such as (iv) equivalent latitude, (v) dynamic tropopause and (vi) polar vortex characterization. The major finding is that PV agrees well between the various data sets on the time scales studied in this work. The authors also highlight the situation where more caution is necessary when working with PV. Some differences between the various data sets arise in particular for (i) equivalent latitude calculations at low latitudes or high altitudes, (ii) the dynamic tropopause in regions of jetstreams and of strong topography, as well as (iii) during the formation and demise of the polar vortex.

This work is intended as part of the S-RIP special issue where it perfectly fits. Such a comparison of PV from different reanalysis data sets has not been completed yet, although PV from reanalysis is a widely used diagnostic to analyze transport and dynamics in the troposphere and stratosphere. The questions asked in the paper are clear. The analysis is very convincing; data and methods are well described. The figures are well structured and clear to understand. The conclusions are based on the analysis. It is easy to follow the thoughts of the authors and as a reader I have the feeling that the authors really know what they are talking about. In general, I think this study will be of great value to users of reanalysis data and as such I would support publication of this study in ACP in the S-RIP special issue. I have some rather minor comments listed below, which the authors might consider for a revised version.

Comments:
• P3, L11: To my knowledge, ERA-I provides relative vorticity, see eg. the ERAInterim data catalogue:
https://apps.ecmwf.int/archive-
catalogue/?stream=oper&levtype=ml&expver=1&month=jan&year=1979&type=an&class=ei
Great catch, thanks! The sentence will now read: "CFSR/CFSv2 provides absolute vorticity while ERA-Interim provides relative vorticity, hence, for …"

• Equation 1: maybe it is worth mentioning that this is the synoptic approximation of PV. We added: "By using the provided or derived vorticity fields, we make use of the synoptic approximation to calculate PV, which assumes that $\varsigma_\theta+f$ approximately equals the absolute vorticity and that horizontal gradients of potential temperature are small."

• Equation 2: I think, since not everybody might be familiar with sPV, some readers would benefit from a comparison of sPV and PV, maybe shown here for one reanalysis but for different averaging times, e.g., a snapshot, monthly, or yearly mean.
We will include the following figure:

[Figure]

Figure 1: January 1st 2005 PV (left) and sPV (right). Note that sPV has similar order of magnitude values throughout the stratosphere as opposed to PV (for which color bar is non-linear).

The text about sPV will changed to: "This scaling is performed to provide fields with a similar order of magnitude throughout the stratosphere as opposed to PV (which increases approximately exponentially with increasing θ, **as shown in Figure 1**)."

• P3, l26: Could you mention the potential temperature range here.  We added in brackets "(330, 340, 360, 380, 400, 420, 440, 460, 480, 500, 520, 540, 560, 580, 600, 620,  660, 700, 750, 800, 850, 900, 960, 1040, 1120, 1200, 1300, 1400, 1500, 1600, 1700, 1800, 1900, 2000, 2100, 2200, 2300, 2400, 2500). "

• P4, l34: Could you say something about the cause of the low bias of sPV? Is this an effect of vertical grid spacing or model physics (e.g. GW drag)?

We do not really know, we did add that in the next figure (the one discussing the discontinuities) that the constant bias at 2500K is likely related to the low lid height for CFSR/CFsV2. This was also mention in the EqL comparison.

• Sec. 4: When there is a 4.1, there should be a 4.2 as well. I would suggest to find a subhead for the first paragraphs of this section.
Subsection 4.2 was converted into a new section with the title "Variations due to differing calculation methods"

• Figure 4 and related discussion: Is Fig. 4 based on the temporal and zonal mean of the entire data set? Can you say something about the order of the magnitude of the differences, if shorter time periods are considered (monthly means or even shorter).

We will include the following figure in the appendix

[Figure]

Caption: Seasonal root-mean-square (RMS) daily (1980-2014) sPV differences between the sPV from each reanalysis provided vorticity or PV and the sPV computed from that reanalyses' horizontal wind, pressure, and temperature fields. Overlaid contours show each reanalyses' climatology based on that reanalyses' provided vorticity.

We will add the following text:

"RMS daily differences arising from different methods of calculating PV are also small (see Figure A2), no larger than 0.3x10-4 s-1 and  mostly better than 0.05x10-4 s-1."

• P6, L8: climatologies "of what" ? Trace gases and aerosols, the sentence will be changed to:  to construct trace gas and aerosol climatologies.

• P6, L13-15: Is there a reference for the EqL computation used here, so that a reader may be able to look up the computation in detail?

We will add:
**"EqL is computed as,**
**$Eql = sin\text{-}1\ (A/2piR^2\ \text{-}1)$**
**where A= A(q) is the area in which PV is less than q on a particular isentropic surface, and R is the radius of the Earth.**
EqL is computed using the 0.5 gridded PV fields using a piecewise constant method, where the PV value is  assumed constant within each grid cell. **Simply, for each PV value, on a given isentropic surface, we sum the areas for all grid cell with smaller field values**. Further, EqL is only …."

• P6, L17: This is potentially the only greater question which I have. Generally, when you speak of variability, here it is talked about the variability along the polar vortex edge, is this a variability caused by the fact that the reanalyses differ in the representation of the atmospheric features among each other, or because there is a large natural variability of the feature. Maybe it could help to also look at the variability of the shown quantities in individual reanalysis data sets to show whether these already have a large variability or not.

The variability in the REM is due to both, mostly due to large natural variability of the feature and differing representations among the reanalysis. Below are the standard deviations for the individual reanalysis for sPV, EqL and the dynamical tropopause. As shown there is large  natural variability of these features in the individual reanalysis.

[Figure]

[Figure]

[Figure]

We will add the following:

In the sPV section: "This variability arises from to a combination of large natural variability with the slightly different representations of sPV among the reanalyses."

In the Eql section: "The largest variability is found along the polar vortex edges, as well as at the top of the upper troposphere subtropical jet in all seasons (likely primarily related to EqL becoming a less appropriate coordinate near / below the tropopause, e.g., Manney et al.,2011; Pan et al., 2012); **that is, in regions of large natural variability in EqL."**

In the tropopause section: "Generally, the differences are within 0.1 km over most of the globe, **except in regions of large natural variability.** Around …"

• P7, L7: Do you search the tropopause from top or bottom or asked differently do you refer here to the lowest or highest tropopause in the presence of multiple tropopause as can occur in the vicinity of tropopause folds?  We added at the end of that paragraph: "Results shown here are for the primary (i.e., lowest) tropopause."

• P7, L29: I wonder whether the parameterizations of orographic GW really have an effect on the tropopause altitude or whether the effect seen here is rather related to larger, resolved GWs themselves above such orography. As far as I know the standard orographic GW parameterizations rather affect higher altitudes by dumping energy at a specified level somewhere in the middle to upper stratosphere and thus affecting the resolved mean flow at those altitudes but not at the tropopause level. Could this here be also a result of the data assimilation, since these are regions with relatively frequent GW occurrences which might be included in radiosonde data which become assimilated?

The reviewer is correct, we do not have enough information to know if the tropopause differences are due to the parametrization differences or due to resolved orographic gravity waves, or differences in assimilated data that may include gravity wave information. We will modify the text simply to: "Other ~1 km discrepancies can be found over Greenland and over the Andes mountains **and are likely related to orographic gravity waves that are common in these regions (e.g., Leutbecherand Volkert, 2000; McLandress et al., 2000; Wu, 2004; Doyle et al., 2005; Fritts et al., 2010).**"

References:
McLandress (2000) - 10.1029/2000JD900097
Leutbecher(2000) - 10.1175/1520-0469(2000)057<3090:TPOMWI>2.0.CO;2
Wu (2004) – 10.1029/2004GL019562
Doyle (2005) - 10.1175/JAS3528.1
Frits(2010) - 10.1029/2010JD013891

In the summary section the text changed to: "where mismatches in the location of the sharp decrease in tropopause altitude from the tropics to mid-latitudes are so common as to affect the climatology; **over Greenland and the Andes regions that are affected by orographic gravity waves;** and over Antarctica, where conventional input data are most sparse"

• In the discussion of Fig. 7 starting on P7, l24, I wonder how much the shown differences could be related to the vertical grid spacing of the individual reanalyses? These data sets all differ in their absolute vertical grid spacing as well as the interpolation may be dependent on the actual location of the individual model levels in the tropopause region. Maybe it would be worth adding information about the vertical grid spacing of the reanalysis products in the tropopause region/stratosphere.

A column was added to Table 1 listing the UTLS vertical spacing around 1.2km for MERRA2 and 1.km for ERA-Interim, CFSR/CFSv2 and JRA55.
We also added the following sentence at the end of the paragraph:  Part of these differences may be due to the slightly different spacing between model levels (1 to 1.2 km apart at these altitudes) and the actual location of such levels with respect to the tropopause.
We also added in Table 1: "the approximate vertical resolutions of the reanalysis fields for their entire vertical range can be found on Figure 3 of Fujiwara et al. (2017)"

• P9, l4: ...smaller THAN the polar cap  Done

• affiliation 2 and 6 are the same  Affiliation 6 was deleted.

**Reviewer 3**

Summary:

In this manuscript, the authors present a thorough intercomparison of reanalysis potential vorticity and diagnostics related to potential vorticity. One modern reanalysis product from four of the major global operational and research centers are selected: CFSR/CFSv2, ERAInterim, JRA-55 and MERRA2. As part of the SPARC Reanalysis Intercomparison Project, this paper provides clear advice to users of reanalysis PV diagnostics as well as recommends reanalysis centers to consider including PV on model levels in future products for optimal comparisons and scientific studies in the future. My one concern is the number (15) and size (multi-panel) of the figures relative to the text, however I do not see any that can be cut down in size or removed from the main text and included in a supplemental instead. Therefore, I recommend this paper for publication after my minor and technical comments below are addressed.

Comments:
Pg 1 Line 8, Pg 2 Line 32: Add "NASA" before "Modern" since the reanalysis center for the other three products is given.   We added NASA as requested

Pg 2 Line 21: I think the comma after "Nash et al., 1996)," should either be a semi colon or a period. We changed the comma to a semicolon

Pg 3 Line 4: Update Table 1 MERRA2 reference to match Gelaro et al. 2017 reference used here. Done

Pg 3 Line 4: Is there a reason ERA-Interim is used instead of ERA-5, the latest ECWMF reanalysis? When we were performing the analysis we did not have access to ERA-5, there were persistent problems downloading the data for an extended time period.

Pg 3 Line 7: Why is the period 1980 through 2014 used? CFSR/CSFv2 was only available on model levels for that time period That is why Long et al 2017 (10.5194/acp-17-14593-2017) or Manney and Hegglin 2018 (10.1175/JCLI-D-17-0303.1), and many others, also used that period.

Pg 3 Line 11, Pg 5 Line 21: I checked the ERA-Interim website and I see "Vorticity (relative)" for ERA-Interim on Model Levels. https://apps.ecmwf.int/datasets/data/interim-fulldaily/levtype=ml/
Yes, we swapped by mistake which field was provided in which reanalysis, the sentence will now read: "CFSR/CFSv2 provides absolute vorticity while ERA-Interim provides relative vorticity, hence, for …"

Pg 3 Line 26: ERA-Interim and JRA-55 have coarser resolution than 0.5x0.5 degree. Could this impact your study?
We used a bilinear interpolation to get to the 0.5°x0.5°grid so the only effects of interpolating to a finer grid should be minimal  (since no extrapolation is involved), further, we did the same analysis at 2degrees resolution and the conclusions are the same.

Pg 4 Line 9, Pg 5 Line 2: Correct me, but I do not see the different processing streams discussed or labelled on Figure 1 or 3, so do we assume that this does not impact PV anomalies? There is mention of a "CFSR to CFSv2 transition" on Pg 8 Line 6 but when this occurred is not stated.

The processing streams will be labeled in what were Figures 1,3,6,8,15 in the original manuscript, that is, in all the timeseries.   The location of the CFSR/CFSv2 transition is mention in section 3.

Pg 4 Line 12: Should "regions" be singular since only the south pole is referenced here and not both poles?
Yes, that is correct. It was changed to region.

Pg 4 Line 32: Can the authors comment on the fact that the CFSR differences are of opposite sign to the other reanalyses.
We will add: "In contrast, the other reanalyses are biased slightly high (only up to 0.3x10-4 s-1) as an artifact of using the REM as a comparison tool. The similarities among these slightly high biases suggest good agreement among ERA-Interim, MERRA2, and JRA-55 at these levels."

Pg 5 Line 11: Can the authors comment on what may cause this difference at CFSR at 2500K? Is this at all related to the model resolution or how the model treats the upper atmospheric levels? or that CFSR has the lowest Lid height (0.26 hPa, Table 1)?

Yes, this may/might/could be related to the lid height, which is right around that level. We will add:  "This may be because this level  is near the CFSR/CFSv2 lid height (0.26 hPa). "

In the EqL section we also added: "… however pronounced differences, greater than 10∘, are seen near the poles around 2500 K, **which may be an artifact caused by the low CFSR/CFsV2 lid height**."

Pg 5 Line 12: "near 850K" looks more like 850-1000K to me. Are the pixels centered on an isentropic level?
No they are not, we will change to: "displays a discontinuity between 800 and 1000K. "

Pg 6 Line 1: UTLS is not defined.   The phrase: "versus those on native model levels, e.g., for Upper Troposphere Lower Stratosphere (UTLS) studies" was added to this sentence.

Pg 6 Line 15: The lowest levels are not of interest to this study but can the authors comment on the "highest levels". Does the top of the atmosphere change between the models that this criterion matters at all? By highest levels we actually meant above 2500K (the top of the model should not affect this except perhaps for CFSR/CFSv2), we will change the sentence to: "This criterion only affects the lowest levels studied in this analysis."

Pg 6 Line 17: It is hard to see along the bottom of the figure. What is the lower limit on Figure 5? Is it 400 K?
Yeah, it is hard to discern, the lower limit is 330K as specified in section 2, which now includes the potential temperatures that we used.

Pg 6 Line 21: Suggest adding "; however" connecting these two sentences.   Done

Pg 6 Line 31-Pg 7 line 2: Can the authors comment on why this might be?   Not really …

Pg 7 Line 4: This looks less evident for MERRA2 after 2005. That is correct, that was covered in the statement before that mentions that most of the time the differences are within 1∘.

Pg 7 Line 19: is altitude above sea level or above surface (ground-level)?   Above sea level, we will change the sentence to: "Figure 8 shows climatological REM dynamical tropopause altitude (**above sea level)** maps for different seasons"

Pg 7 Line 26: "can be up to 1 km", does this have anything to do with the difference model resolution in the UTLS?
Yes it can be related to it. A column was added to Table 1 listing the UTLS vertical spacing around 1.2km for MERRA2 and 1.km for ERA-Interim, CFSR/CFSv2 and JRA55. We also added the following sentence at the end of the paragraph: "Part of these differences may be due to the slightly different spacing between model levels (1 to 1.2 km apart at these altitudes) and the actual location of such levels with respect to the tropopause."
We also added in Table 1: "the approximate vertical resolutions of the reanalysis fields for their entire vertical range can be found on Figure 3 of Fujiwara et al. (2017)"

Pg 8 Line 12: add "(not shown)"  Added

Pg 8 Line 14: In general for this section, do the authors use the native resolution or the 0.5x0.5 degree interpolated resolution?
The sPV thresholds were computed on their native resolution, we change the text to: "we bin sPV **(from the native resolution of each reanalysis)** as a function of equivalent latitude, differentiate,…"

The rest was done using the 0.5x0.5 fields, we changed the text to: "To quantify such differences we identify vortices for each day and catalog the number of vortices as well as their area. To identify the vortices on a given isentropic surface (i**n the 0.5◦by 0.5◦ gridded fields),** we use a flood filling …"

Pg 9 Lines 30-31: I suggest referencing Figure 11 here since later in this paragraph you reference Figure 12.  The sentence was changed to: "In midwinter (see Figure 12), maximum differences .. "
Note that following a request by reviewer 2 we added a new figure hence the change in Figure number.

Pg 10 Line 3: Is there a reference for selecting this vortex area of 0.15*10^7 km^2?   Not really this threshold was chosen somewhat arbitrarily by looking at the behavior of the plots.

Pg 10 Line 17: capitalize "southern"   Done

Pg 10 Line 20: Looks to me CFSR is at 500K and 440K.   Yes we will change the text to: "with the exception of CFSR/CFSv2 at 440 K (and to a lesser degree at 500 K), which shows a clear departure…"

Pg 10 Line 22: Suggest moving the sentence starting with "MERRA-2" to after "midwinter)." to keep the upper-level discussion together.   The paragraph was changed to: "The greatest variability among the equivalent ellipses is seen at 1100 K and 1300 K, consistent with the variability in area seasonality (up to 20% in midwinter), with MERRA-2 showing slightly smaller ellipses than the other reanalyses at 1300 K. Most of the reanalyses agree remarkably …"

Pg 10 Line 26: add "both" before "showing" Done

Pg 10 Line 26-27: Could the discontinuity in ERA-I be related to a change in processing streams?

The updated figure includes the processing streams; this discontinuity is not related to either data changes or processing streams.

Figures:

Figure 3: The y-axis has minor ticks which seem to be greater than the resolution of the pixels. I recommend reducing the minor ticks. Can annual minor ticks be added to the x-axis on this and the other figures?
The number of y-minor ticks was reduced as requested and we added annual x-minor ticks for figures 4 7,9,16    (that is, figures 3, 6, 8 and 15 in the previous manuscript)

Figure 7: Can the y-axis include 30degree latitude interval labels since it is referenced several times on Page 7.  We added a dashed line at 30S and 30N in every panel. The caption will be updated to include: "Dashed lines indicate the 30S and 30N latitudes."